# Spatial Pattern of Species Richness among Terrestrial Mammals in China

**Yao Chi [1] , Jiechen Wang [1,2,\*], Changbai Xi [1], Tianlu Qian [1] and Caiying Sheng [1]**

[1] Jiangsu Provincial Key Laboratory of Geographic Information Science and Technology, Key Laboratory for Land Satellite Remote Sensing Applications of Ministry of Natural Resources, School of Geography and Ocean Science, Nanjing University, Nanjing 210023, China; chiyao@smail.nju.edu.cn (Y.C.); xicb11@smail.nju.edu.cn (C.X.); tianluqian@outlook.com (T.Q.); shengcy@smail.nju.edu.cn (C.S.)

[2] Jiangsu Center for Collaborative Innovation in Geographical Information Resource Development and Application, Nanjing University, Nanjing 210023, China

\* Correspondence: wangjiechen@nju.edu.cn; Tel.: +86-025-89680669

**Abstract:** We describe large-scale patterns of terrestrial mammal distribution in China by using geographical information system (GIS) spatial analysis. Mammal taxa, examined by species, family, and order, were binned into 10 km × 10 km grids to explore the relationship between their spatial distribution and geographical factors potentially affecting the same. The spatial pattern of species richness revealed four agglomerations: high richness in the south, low in north, and two low richness areas in eastern and western China. Species richness patterns in Carnivora was the most similar to overall terrestrial mammals' richness; however, species richness in different orders exhibited distributions distinct from the overall pattern. We found a negative relationship between richness and latitude gradient. Species richness was most strongly correlated with forested ecosystems, and was found to be higher at an elevation of 2000~2200 m, with greater altitudinal variation indicative of higher species richness.

**Keywords:** GIS; spatial patterns; species richness; terrestrial mammals

## 1. Introduction

China is a major biodiversity hotspot, with over 7300 vertebrate taxa (11% of the world's extant species). Unfortunately, the rapid acceleration of anthropogenic activity poses a severe threat to biodiversity [1], with the number of endangered species far outstripping available conservation resources. In particular, the negative consequences stemming from human activity strongly affect mammals, given their wide-ranging distribution in nearly every habitat [2–4]. Additionally, 178 (26.4%) of 673 mammal species of China were classified as regionally threatened during the 2015 Redlist assessment [5]. Therefore, having an accurate, up-to-date understanding of mammalian distributional patterns, not only at a local, but also at a regional level, is important for conservation efforts and for promoting the sustainable development of human society.

Measuring biological diversity is an interdisciplinary task that combines ecology and conservation biology [6]. Numerous studies have investigated the diversity of groups across the animal kingdom [7–11]. Some of these studies emphasized species richness and evenness [12,13], while others also included data on evolutionary history and species function within ecosystems [14–16]. Typically, such works rely on field observation and are limited to smaller study areas [17–19]. Statistical models are also frequently used to analyze and predict distribution [20,21], focusing on the analysis of the single species distribution characteristics [22–25].

In China, most research explores species-distribution patterns limited to small administrative units [26,27], increasing the difficulty of comparing regions of different areas. Few studies have

examined large-scale geographical distribution of terrestrial mammals in China because of a lack of macro-level data. Xie et al. [28] published a wonderful analysis of China's biogeographical regions and tested it with a large subset of mammalian species distribution data. Later, Smith and Xie [29] revisited and expanded this portrayal of mammals in China. Furthermore, previous reports tended to use relatively coarse spatial grain, typically at around 10,000 km$^2$ [11,30]. In keeping with this relative lack of raw geographical data, most previous research concentrated on mathematical statistical models [31–33], rather than including empirical information on spatial structure. However, as geographical information system (GIS) technology continues to influence research on animal distributions [34,35], we have seen powerful spatial analyses applied to biological-resource management, animal-habitat evaluation [36,37], species-distribution predictions [38–40], and biodiversity conservation [41–43].

There are too few field studies being conducted in China to inform current range maps for most species. In this study, we examined the distribution of terrestrial mammalian species, families, and orders by combining GIS analysis with recent taxonomic data on China's mammal diversity and geographic distribution [3]. In particular, we wished to assess patterns of distribution and focus on species abundances-both high and low-and their distribution throughout China. We also analyzed conservation status of mammals and distribution of threatened species. In addition, we included high-precision spatial data on terrain and vegetation types. Our analysis should provide a clearer picture of macro-distribution patterns among China's terrestrial mammals, thus contributing to biodiversity conservation in this region.

## 2. Materials and Methods

### 2.1. Study Area

The study encompassed all of China's land mass (9.6 million km$^2$), including Taiwan and Hainan Island but excluding the surrounding ocean and any islets (Figure 1; 73°29′ E–135°2′ E, 3°31′ N–53°33′ N). The country is characterized by complex terrain with diverse landforms, including plateaus, mountains, plains, hills, and basins. Climate varies widely from the tropics to the cool-temperate zone, with a clear division between dry and wet regions. Vegetation also is diverse, covering a wide range of zones, including rainforest, steppes, and desert. This overall diversity of habitat has resulted in a high number of species. The mammal fauna, for instance, belongs to both Palearctic and Oriental Regions [44], large groups that can be further subdivided into seven biogeographic subregions (Northeast China, North China, Inner Mongolia-Xinjiang Region, Qinghai-Tibet region, Southwest China, Central China, and South China) [45].

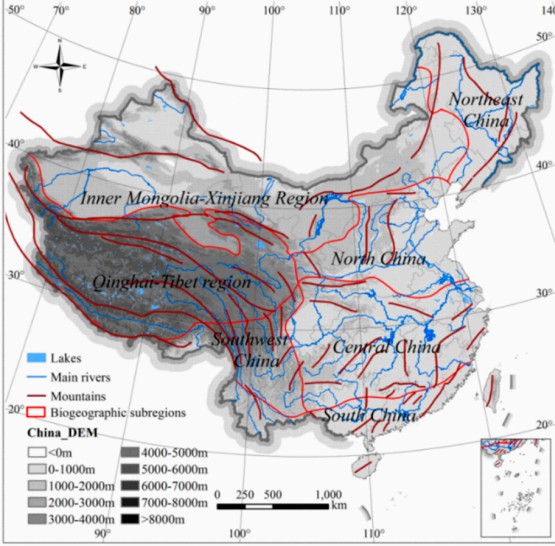

**Figure 1.** Topography and regional zones of China.

### 2.2. Data Resources

In terms of distribution data among animals, GBIF database have provided relatively abundant observation data with a long time span. However, there is no systematic collection of mammalian distribution data in China, especially for endangered species. In the study, mammal species' distribution ranges data were obtained from China's mammal diversity and geographic distribution [3], which is more systematic, organized by many zoologists. Jiang et al. (2015) used A Completed Checklist of Mammal species and subspecies in China: A Taxonomic and Geographic Reference by Wang Yingxiang (2003) and Mammal Species of the World (3rd Edition) by Wilson and Reeder (2005) as templates, collected all available relevant information of books and literatures on taxonomy, and distribution of mammals in China by the end of March of 2015. After five evaluation meetings and two rounds of evaluation by correspondence, with input from about 80 mammalogists in China, they finalized The Checklist of China Mammal Diversity. Thus, the most updated information on China's mammal diversity is presented in this book, the distribution of each species is plotted. The species distribution maps are compiled from national and regional atlases and books, as well as on species observations of field work.

The database in Jiang et al. [3] contains 673 species representing 12 orders, 55 families, and 245 genera occurring in China. Our analysis excludes 1) humans (Homo sapiens); 2) marine carnivores (northern fur seal—Callorhinus ursinus, Steller sea lion—Eumetopias jubatus, and three seal species); all marine Cetartiodactyla; the dugong (Dugong dugong); the Senkaku mole (Mogera uchidai); and three locally extinct species from Yunnan (Sumatran rhinoceros—Dicerorhinus sumatrensis, Javan rhinoceros—Rhinoceros sondaicus, and Indian rhinoceros—R. unicornis). Thus, our study included 624 species in 11 orders and 42 families.

Species distribution maps were uniformly produce in Asia Lambert Conformal Conic projections. Distributions of different mammal species were represented as polygon features that overlapped with each other in space. The analysis explored spatial distribution on a mesh scale to eliminate the effect of area on species richness. The entire study area was divided into 10 km × 10 km grids (96,802 grids total). Distribution data were converted to vectors based on the grids; this format does not change the distribution, but simplifies data processing.

The analysis also included DEM (Digital Elevation Model, SRTM 30 m) data and terrestrial ecosystems in China. To classify topographic fluctuation [46,47], we defined altitudinal amplitude by window analysis of 5 km × 5 km grids extracted maximum and minimum elevation data through resampling of 1 km × 1 km grids, yielding the maximum difference between each grid elevation. The macro structure data of China's land ecosystems are based on the data of 1: 100,000 land use/land cover obtained by remote sensing interpretation. Through the identification and research of each ecosystem type, the spatial distribution datasets of 1: 100,000 land ecosystem types in China was formed after classification processing. The data set is provided by Data Center for Resources and Environmental Sciences, Chinese Academy of Sciences (http://www.resdc.cn). Ecosystem type has 7 categories, including farmland ecosystems, forest ecosystems, grassland ecosystems, water and wetland ecosystems, desert ecosystems, settlement ecosystems, and other ecosystems. These data are preprocessed before putting into use, including clipping and transforming uniform Asia Lambert Conformal Conic projection.

### 2.3. Analysis of Taxonomic Spatial Distribution

Multi-layer overlay statistics of GIS in 10 km × 10 km grids were superimposed over distributions of the 624 terrestrial mammal species. Mammal presence or absence in 10 by 10 km block was recorded as "1" and "0," respectively. The number of mammalian species, families, and orders per grid were counted to determine the richness indicators of "species", "families", and "orders" within each taxonomic category.

Exploratory Spatial Data Analysis (ESDA) can be defined as the collection of spatial data analysis methods and techniques to describe and visualize spatial distributions of objects or phenomena, identify patterns of spatial agglomeration and spatial outliers, and reveal the mechanism of spatial interaction among research objects [48]. Central to ESDA is the concept of spatial autocorrelation, including global and local spatial autocorrelation methods. The resultant richness value was spatially weighted using the method of polygon contiguity edges corners, then combined with the global Moran's I index [49] and the Local Getis-Ord Gi* statistics [50] in the GIS geostatistical method. The ESDA method was used to explore differences in the spatial aggregation of species and family richness. First, the analysis employed global Moran's I, a commonly used spatial autocorrelation index [49] that evaluates departure from randomness, with a significant value indicating a non–random distribution. We calculated Moran's I from richness indices and improved robustness through 999 randomization operations. Local Getis-Ord Gi* statistics were also used to identify the area of an aggregation, or hot spots in distributions. This parameter accurately detects clusters of hot spots in overlapping range data and reveals the distribution of these hot spots.

In order to quantitatively measure the relationship between richness and other variables, correlation analysis were used to measure the influence of specific factors (Latitude, Elevation, Ecosystem type) on species richness. It can be represented as [51]:

$$R\,(A, B) = \pm\max\Big\{\big|\,rl\,(A, B)\,\big|, \big|\,rnl\,(A, B)\,\big|\Big\} \tag{1}$$

$$rnl\,(A, B) = \pm\max\Big\{\big|\,rl\,(\ln A, B)\,\big|, \big|\,rl\,(A, \ln B)\,\big|, \big|\,rl\,(\ln A, \ln B)\big|\Big\} \tag{2}$$

where $R\,(A,B)$ represents the connection coefficient to explore the relationship of the two variables, which is the larger value between the linear correlation coefficient of the two variables (Pearson's correlation coefficient, $r_l\,(A,B)$) and the nonlinear correlation coefficient ($r_{nl}\,(A,B)$). The larger value among the Pearson's correlation coefficient of the natural logarithm of *A* and *B*, *A* and the natural logarithm of *B*, and the natural logarithm between *A* and *B* is taken as the $r_{nl}\,(A,B)$.

## 3. Results and Analysis

### 3.1. Spatial Pattern of Terrestrial Mammals Total Group Richness

To analyze the spatial distribution of mammals, 10 by 10 km blocks occupied by terrestrial mammalian richness were tallied to detect the distributional characteristics. Densities of overlapping species ranges vary throughout the country (Figure 2), although in general terms, the south has significantly higher richness than the north. Richness patterns across distinct taxonomic category had similar spatial distribution patterns overall (Figure 2a,b). Within each taxonomic category, the distribution region exhibited some overlap, whereas small differences were present across taxonomic levels. Species richness and family richness were highest in the southern part of Central China, Southwest China, and South China, especially the Yunnan region. However, family richness was obviously distributed in the southeast, but species richness was not. Across northwestern and northeastern China, as well as Inner Mongolia-Xinjiang, the range of family-level distribution clearly expanded beyond species-level.

Based on spatial distribution data, the Hengduan Mountains in Southwest China region has the most terrestrial mammals. In contrast, eastern and northwest China are low in species richness. Mammal richness was lowest, with few species in the Qinghai-Tibetan Plateau of western China. Spatially, abundance of species was found to be high in the south but low in the east and west.

The ESDA method was used to explore differences in the spatial aggregation of species and family richness. Moran's I indices were greater than 0.98 (P < 0.01) for both species, and family richness, indicating strong spatial dependence between the richness distribution of these two taxonomic categories in China. The spatial congregations of "species" and "family" richness of terrestrial mammals in China are congruent with each other based Local Getis-Ord Gi* statistics (Figure 3). Higher richness

regions are located in the south, while both east and west show two low richness areas. From the view of zoogeographic divisions, the significantly higher aggregation areas of "species" and "family" richness are mainly concentrated in Southwest, Central, and South China. These regions are characterized by abundant richness of species and family. The Qinghai-Tibet region is the largest area with significantly low value aggregation, forming the low-value subsidence area of mammal richness, where the mammal species richness are low. On both sides of east and west in North China, there are significant low value aggregation areas and significant high value aggregation areas, respectively; the heterogeneity in the region is obvious, and the species richness in the western region is obviously more than that in the eastern coastal area. There also are large areas of significantly low richness aggregation in the Inner Mongolia-Xinjiang Region, where the mammal species richness are not high. There was no significant spatial aggregation effect in Northeast China.

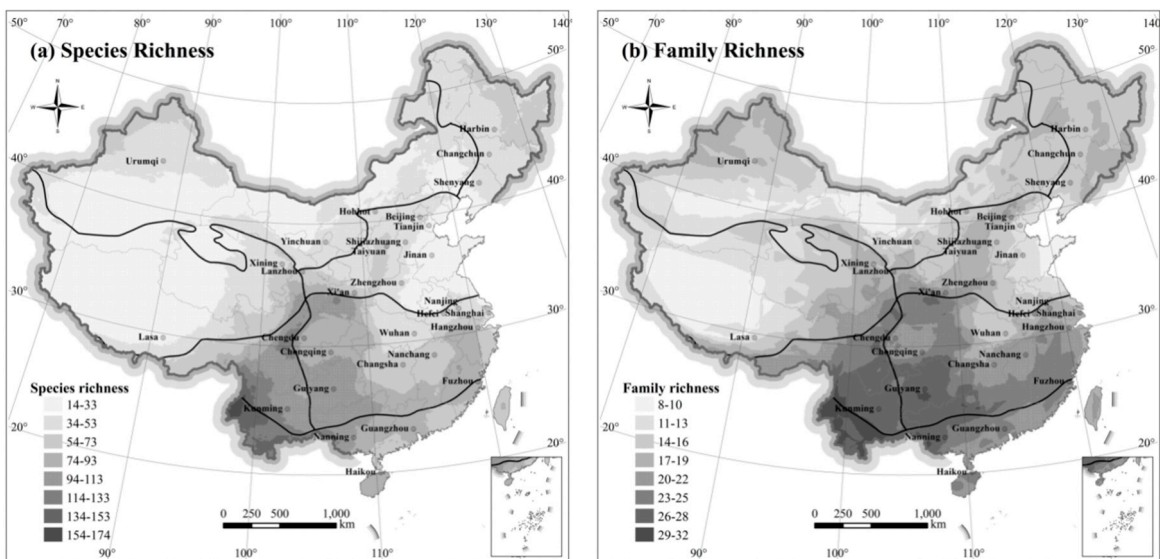

**Figure 2.** Spatial distribution of mammal species (**a**) and family (**b**) richness in China.

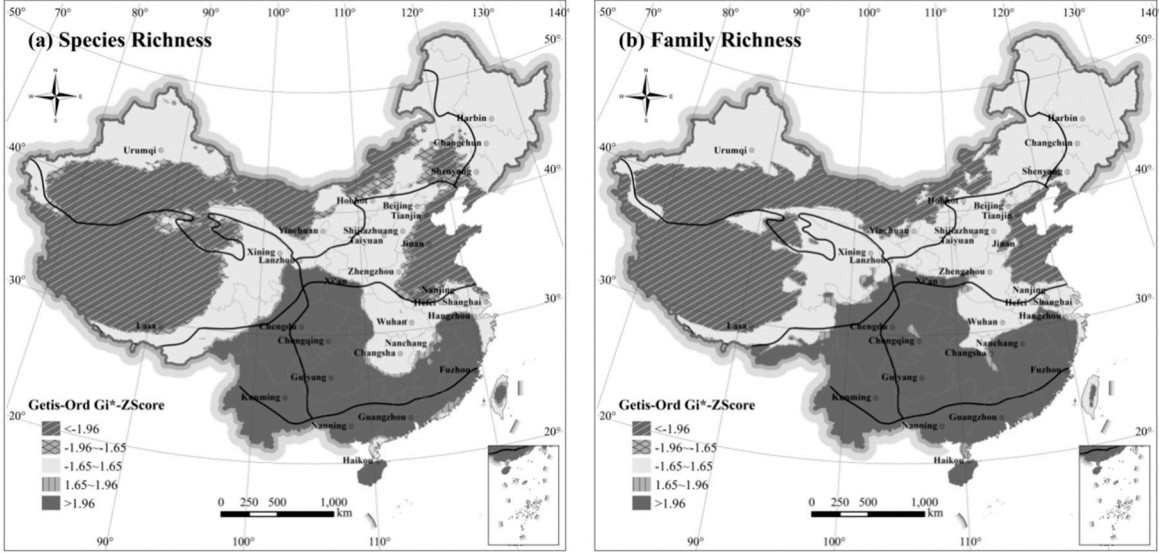

**Figure 3.** Spatial agglomeration characteristics of mammalian richness in China. (**a**) Local Getis-Ord Gi* of species richness; (**b**) Local Getis-Ord Gi* of family richness.

## 3.2. Spatial Distribution of Different Terrestrial Mammalian Orders

There are 11 orders of mammals assessed separately in the present study (see Table 1). Mammals in the orders Perissodactyla, Pholidota, Scandentia, and Proboscidea are few and sparsely distributed in China. Pholidota, Scandentia, and Proboscidea were primarily distributed in South China, as well as the south of Southwest and Central China, and rarely in the north of Qinling Mountains-Huaihe River; their distribution was most concentrated in Yunnan, Guangxi, and other areas containing tropical rainforest habitats. Species in the order Perissodactyla are primarily distributed in the Inner Mongolia-Xinjiang and Qinghai-Tibetan regions, where they are adapted to the xeric open steppe and grassland environments.

**Table 1.** Taxonomic distribution of China's 624 of terrestrial mammal species.

| Order | Number of Species | Proportion of Research Units Occupied | Maximum Species Number Per Unit | Minimum Species Number Per Unit |
|---|---|---|---|---|
| Carnivora | 53 | 100.00% | 32 | 2 |
| Rodentia | 215 | 100.00% | 55 | 2 |
| Cetartiodactyla | 67 | 100.00% | 15 | 1 |
| Lagomorpha | 35 | 96.66% | 11 | 1 |
| Chiroptera | 134 | 85.84% | 39 | 1 |
| Eulipotyphla | 86 | 84.04% | 28 | 1 |
| Primates | 26 | 34.17% | 9 | 1 |
| Perissodactyla | 3 | 24.08% | 2 | 1 |
| Pholidota | 3 | 19.85% | 2 | 1 |
| Scandentia | 1 | 9.86% | 1 | 1 |
| Proboscidea | 1 | 0.42% | 1 | 1 |

The species richness of Carnivores paralleled that of all mammal species across China (Figure 4a). Richness was highest in the south (Southwest, Central, and South China regions), and lowest in the North China, Qinghai-Tibet, and Inner Mongolia-Xinjiang regions.

Species richness of rodents was highest in the Southwest China region (Figure 4b). Rodent species richness was also high in the Tianshan and Altai Mountains located in the northwest part of the Inner Mongolia-Xinjiang Region. Species richness of rodents was lowest in open plains areas across China, such as the Tarim Basin (Inner Mongolia-Xinjiang Region), Qinghai-Tibetan Plateau Region, and xeric areas of North China and Northwest China regions.

Ungulates in the Cetartiodactylla (Figure 4c) are found across China, but are particularly abundant in the Southwest China Region (Hengduan Mountains and eastern Himalaya). Several species are adapted to life on open steppe and alpine meadow environments, such as the Tibetan antelope (chiru—Pantholops hodgsonii), wild yak (Bos grunniens) of the Qinghai-Tibetan Region.

Lagomorphs (Figure 4d), comprised of the families Leporidae (hares, genus Lepus) and Ochotonidae (pikas, genus Ochotona), constitute the most different distribution across China of any order of mammal. The hares (10 species) are fairly distributed evenly across China, whereas the pikas (25 species) have their highest species richness worldwide on the Qinghai-Tibetan plateau.

Chiroptera are the second most abundant group of mammals after Rodentia, extremely widespread (Figure 4e), mostly concentrated in the tropical and subtropical forests. Species richness was relatively high in the Southwest China region and surrounding areas. Species richness was low in most regions in the north. The distribution patterns are the same as that of the terrestrial mammal, a significant spatial heterogeneity that east and west is lower than that of the south.

Eulipotyphla were distributed throughout most of the country (Figure 4f). The higher species richness was concentrated in Southwest China, while species richness in Inner Mongolia-Xinjiang Region was low.

The distribution of Primate species (Figure 4g) was limited in South, Central, and Southwest China, which contain tropical and subtropical habitats. The regions with the highest richness of species in this order ranged from the central and southern borders of Yunnan to the south of Guangxi. No species from this order were found in the north of Qinling Mountains-Huaihe River and Qinghai-Tibetan Plateau.

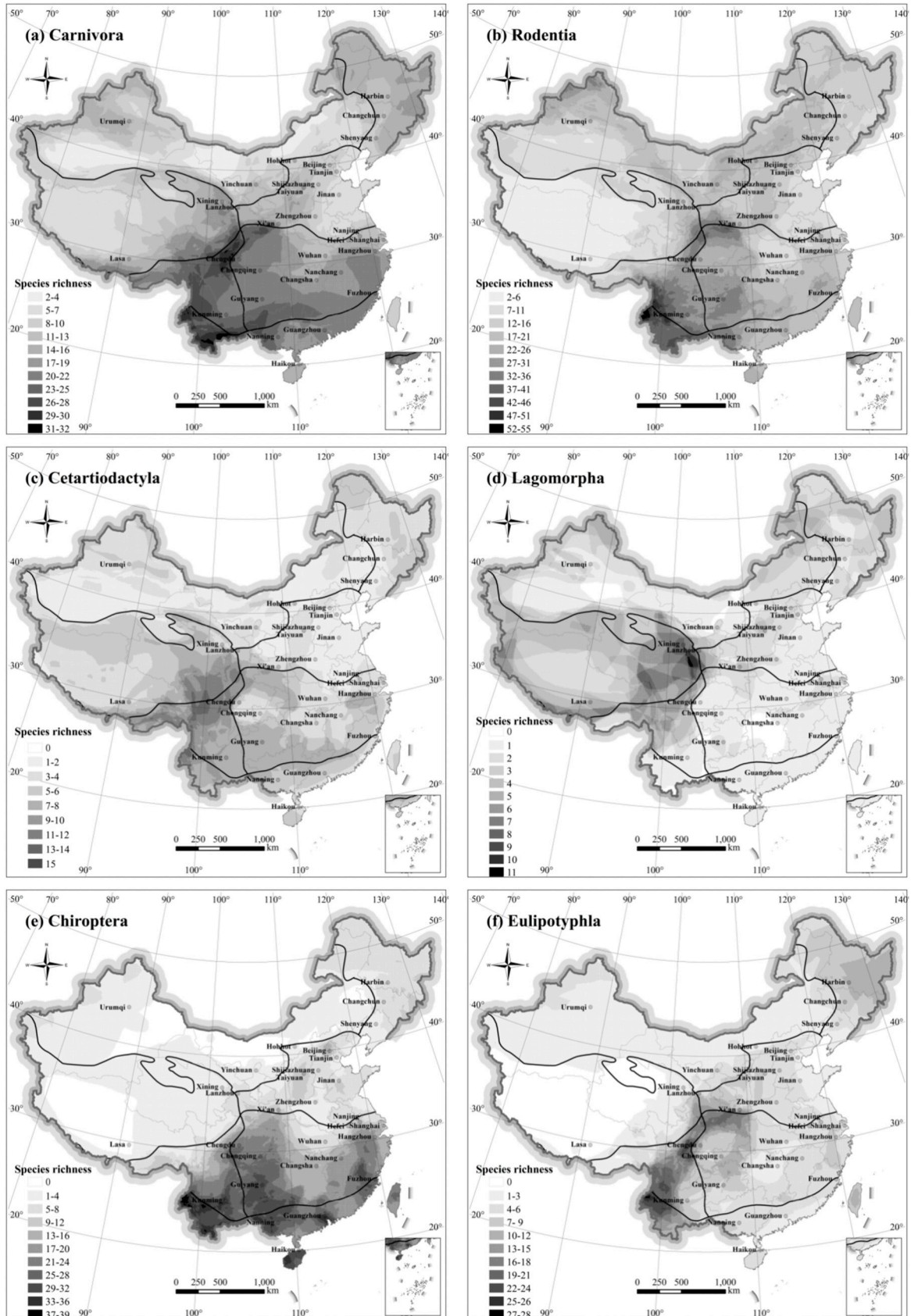

**Figure 4.** *Cont.*

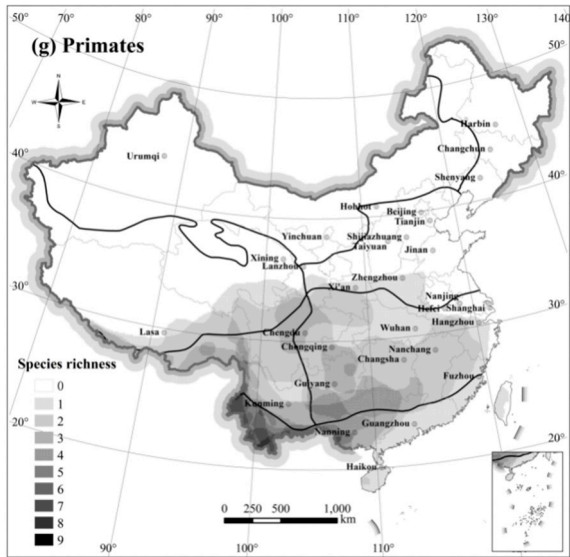

**Figure 4.** Spatial distribution of major mammalian order richness. (**a**) Carnivora; (**b**) Rodentia; (**c**) Cetartiodactyla; (**d**) Lagomorpha; (**e**) Chiroptera; (**f**) Eulipotyphla; (**g**) Primates.

In summary, species richness distribution patterns of different mammal orders had distinct characteristics with, however, overlapping regions and some similarities. All orders exhibited higher species richness in Southwest China, with the exception of Lagomorpha. Species richness was particularly high in the Hengduan Mountains and in south-central Yunnan, but was low for many taxa in the North China Plain, the Tarim Basin, and the hinterland of the Qinghai-Tibetan Plateau.

### 3.3. Level of Threat to Species

In this study, 167 (26.76%) of 624 terrestrial mammals of China were assessed as threatened at the national level [5] (Table 2). Of these, three taxa are evaluated as Extinct in the wild: *Equus ferus*, *Bubalus arnee*, and *Saiga tatarica*. Although *Equus ferus* and *Saiga tatarica* were reintroduced from abroad, captive populations established there, they still need to be artificially raised and bred [5]. Additionally, 167 other species (26.76%) are threatened, including Critically Endangered (55 species), Endangered (48 species), and Vulnerable (64 species). Critically Endangered species widely distributed South China, concentrated in Yunnan province (Figure 5a). Endangered species (Figure 5b) were mainly in the east of Qinghai-Tibetan Plateau. Vulnerable species (Figure 5c) were found across China, but mainly distributed South China.

**Table 2.** Regional Red List status of different mammalian orders of China. The categories are Extinct in the wild (EW), Critically Endangered (CR), Endangered (EN), Vulnerable (VU), Near Threatened (NT), Least Concern (LC), and Data Deficient (DD).

| Order | EW | CR | EN | VU | NT | LC | DD |
|---|---|---|---|---|---|---|---|
| Carnivora | 0 | 7 | 19 | 13 | 13 | 1 | 0 |
| Rodentia | 0 | 1 | 3 | 7 | 36 | 147 | 21 |
| Cetartiodactyla | 2 | 29 | 14 | 11 | 7 | 3 | 1 |
| Lagomorpha | 0 | 2 | 2 | 1 | 8 | 21 | 1 |
| Chiroptera | 0 | 0 | 3 | 15 | 51 | 42 | 23 |
| Eulipotyphla | 0 | 1 | 0 | 11 | 36 | 30 | 8 |
| Primates | 0 | 14 | 6 | 5 | 0 | 1 | 0 |
| Perissodactyla | 1 | 0 | 0 | 1 | 1 | 0 | 0 |
| Pholidota | 0 | 1 | 0 | 0 | 0 | 0 | 2 |
| Scandentia | 0 | 0 | 0 | 0 | 0 | 1 | 0 |
| Proboscidea | 0 | 0 | 1 | 0 | 0 | 0 | 0 |
| Total | 3 | 55 | 48 | 64 | 152 | 246 | 56 |

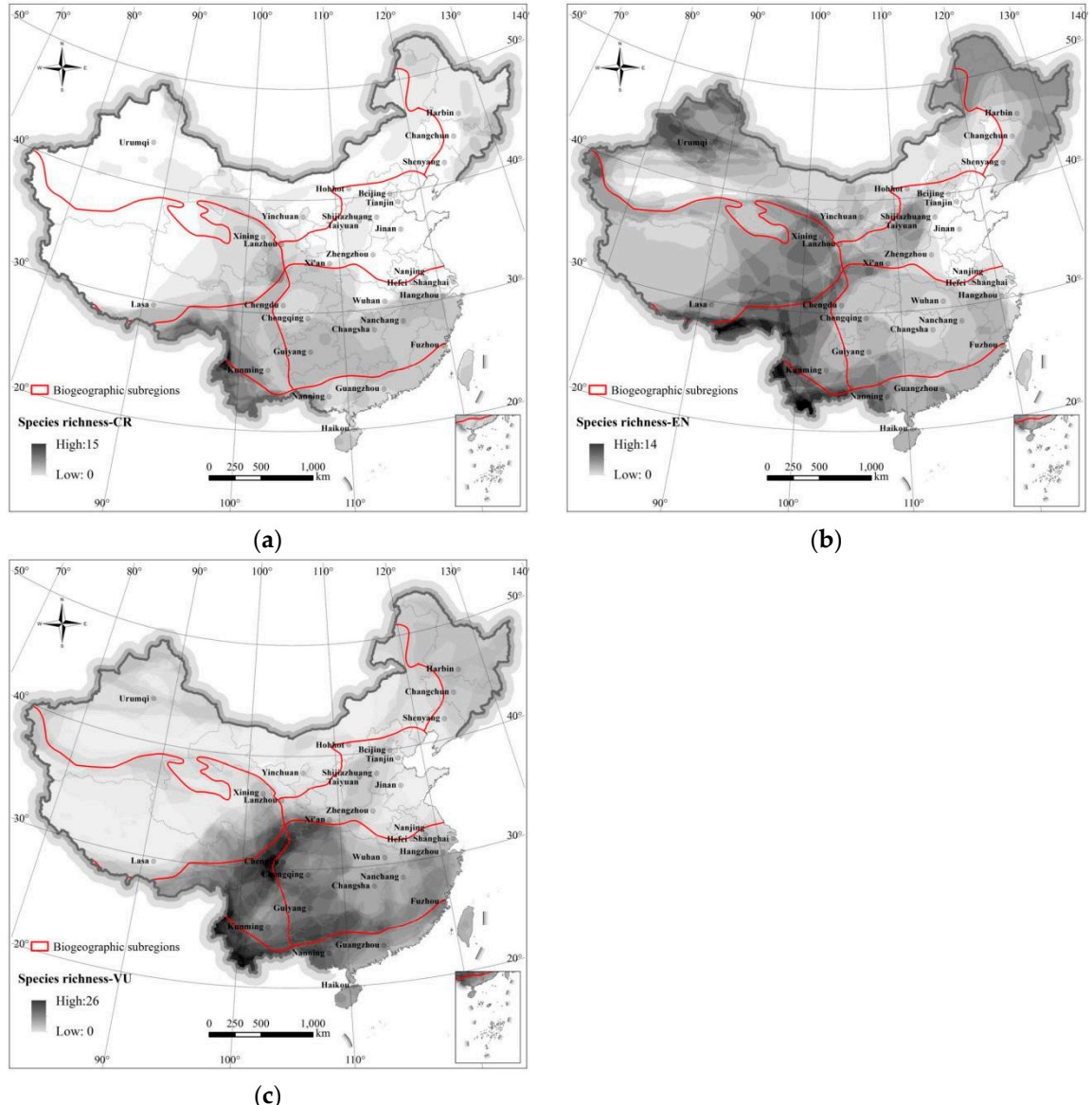

**Figure 5.** Spatial distribution of threatened species. (**a**) Critically Endangered species; (**b**) Endangered species; (**c**) Vulnerable species.

A further 152 species (24.36%) were found to be Near Threatened. A large number of species (246 species, 39.42%) are evaluated as Least Concern, and 8.97% (56 species) as Data Deficient. The order Cetartiodactyla includes the most threatened species (*n* = 54), followed by Carnivora (*n* = 39), Primates (*n* = 25), Chiroptera (*n* = 18), Eulipotyphla (*n* = 12), Rodentia (*n* = 11), and Lagomorpha (*n* = 5), Pholidota (*n* = 1), Proboscidea (*n* = 1), Perissodactyla (*n* = 1; Table 2). In total, more than half (51.12%) of the terrestrial mammals of China have been placed in threatened (Critically Endangered, Endangered, or Vulnerable) or near-threatened categories.

*3.4. Factors Affecting Terrestrial Mammalian Distribution in China*

3.4.1. Latitude

Multiple studies have described the relationship between species diversity and latitude [52,53]. Here, we verified that the distribution of terrestrial mammals in China followed the latitude species gradient: levels in richness of terrestrial mammals in China tend to decrease with increasing latitude (Figure 2). Terrestrial mammal richness at "species" and "family" level is significantly negatively

correlated with latitude (Table 3; Correlation coefficient R of −0.572 and −0.601, respectively), further confirming correspondence with the latitudinal species gradients for terrestrial mammals in China. The distribution of species varies and is latitude dependent. The hypotheses proposed to account for the latitudinal gradient pattern of species richness, including geographic area hypothesis, productivity hypothesis, ambient energy hypothesis, rapoport-rescue hypothesis, evolutionary speed hypothesis, geometric constraints hypothesis, and so on [53]. With the exception of Lagomorpha, other orders show negative correlation with latitude, in particular Carnivora, Cetartiodactyla, Chiroptera, and Primates. It is related to Lagomorph living at a high latitude or high altitude area.

**Table 3.** Correlation coefficients R for comparisons between mammalian richness and latitude.

|  | Correlation with Latitude |
|---|---|
| Species | −0.572 ** |
| Family | −0.601 ** |
| Carnivora | −0.570 ** |
| Rodentia | −0.306 ** |
| Cetartiodactyla | −0.501 ** |
| Lagomorpha | 0.235 ** |
| Chiroptera | −0.708 ** |
| Eulipotyphla | −0.271 ** |
| Primates | −0.716 ** |

** $p < 0.01$ (bilateral). L is expressed as linear correlation coefficient, otherwise it is nonlinear correlation coefficient.

### 3.4.2. Elevation and Altitudinal Amplitude

It has been noted that species diversity has multiple distribution patterns along elevation gradient [54–56]. Terrestrial mammal richness in China is negatively correlated with elevation, with $r = -0.349^{**}$ and $-0.333^{**}$ for species and families, respectively (Table 4). Most individual higher taxa also are negatively correlated with the elevation, whereas Cetartiodactyla, Lagomorpha, and Primates, are more abundant at high elevation.

**Table 4.** Correlation coefficients between mammalian richness and elevation.

|  | Correlation with Elevation | Correlation with Altitudinal Amplitude |
|---|---|---|
| Species | −0.349 ** | 0.387 ** |
| Family | −0.333 ** | 0.380 ** |
| Carnivora | −0.112 ** L | 0.490 ** |
| Rodentia | −0.606 ** | 0.189 ** |
| Cetartiodactyla | 0.357 ** | 0.620 ** |
| Lagomorpha | 0.638 ** | 0.352 ** L |
| Chiroptera | −0.462 ** | 0.253 ** |
| Eulipotyphla | −0.330 ** L | 0.271 ** |
| Primates | 0.254 ** | 0.389 ** L |

** $p < 0.01$ (bilateral). L–expressed as linear correlation coefficient, otherwise it is nonlinear correlation coefficient.

Mammalian richness was lower at very high altitudes (>5000 m), where there may be many endemic species (Figure 6a). The richness is relatively not high in low altitude areas (800~1500 m) and high-altitude areas (4400~5000 m). The richness of middle altitude areas (1800~2600 m) is relatively high, of which the middle altitude of 2000~2200 m are the highest species richness areas [57,58]. Areas of low richness are mainly located in eastern and western China. The latter is dominated by the Qinghai-Tibetan Plateau, a high-altitude area with an average elevation of over 4000 m. The North China Plain in eastern China, with plain and hilly areas, is a low-altitude area, most of which is below 50 m, where the terrain is flat, and the total species richness is still relatively low.

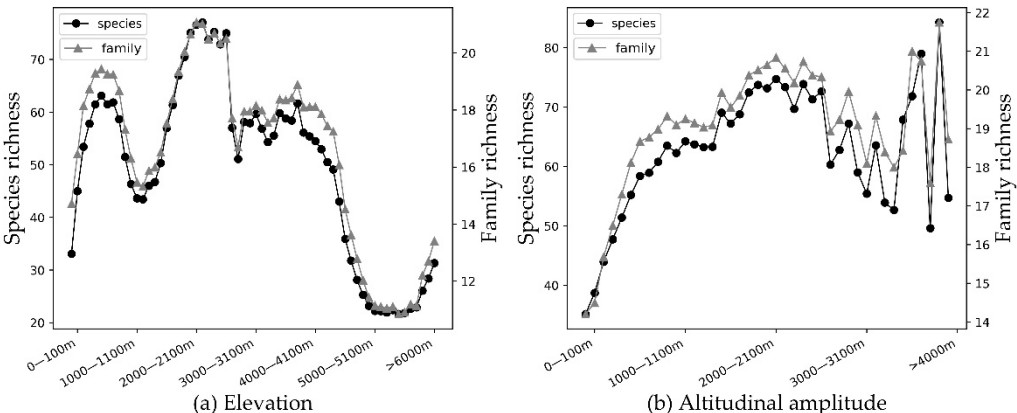

(a) Elevation            (b) Altitudinal amplitude

**Figure 6.** The relationship between average values of terrestrial mammal richness and elevation (**a**), and altitudinal amplitude (**b**).

Terrestrial mammal richness of different taxa in "species" and "family" are weakly positively correlated with the altitudinal amplitude (see Table 4), $r = 0.387$ ** and 0.380 **, respectively. Species richness shows an overall increasing trend with increase in altitudinal amplitude (Figure 6b). Areas with high terrestrial mammal richness in China are mainly concentrated in Southwest China, Central China, and South China (Figure 2). Yunnan Province in southwest China, especially the southwest part of the Yunnan-Guizhou Plateau, has abundant terrestrial mammal richness (see Figure 7).

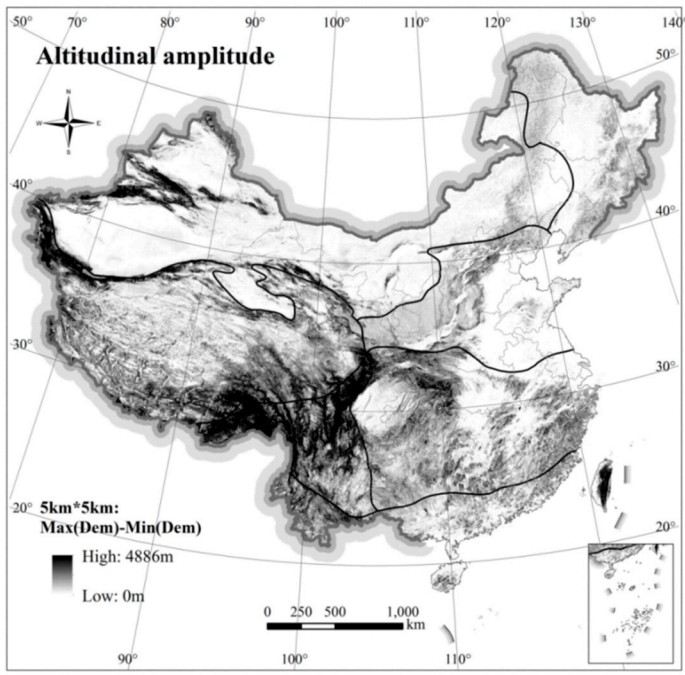

**Figure 7.** Spatial distribution of altitudinal amplitude in China.

### 3.4.3. Ecosystem Type

The relationship between species richness and ecosystem type all showed statistically significant correlation coefficients (Table 5). The richness distribution of terrestrial mammals in China is closely correlated to forested ecosystem. Forest ecosystems in China are primarily distributed in humid areas, and mammal richness is relatively higher in these forested zones. The average number of species in farmland ecosystems is also relatively high. Farmland ecosystems are often embedded with natural ecosystems, including forest, grassland, wetlands, and other regions. The correlation between grasslands ecosystem and mammal richness distribution is not high, and grassland ecosystems are

mostly distributed in arid areas, resulting in lower species richness. There is a significant negative correlation between desert ecosystem and mammal distribution, and the average species number in the studied unit are lower. Desert ecosystems are mostly distributed in arid regions, the species richness in desert areas are extremely poor.

**Table 5.** Correlation between mammalian richness and ecosystem types, and average number of mammals in 100 km² blocks covered by ecosystem types.

|  | Correlation with Species Richness | Correlation with Family Richness | Average Number of Mammals in 100 km² Blocks Covered |
|---|---|---|---|
| Farmland ecosystem | 0.473 ** L | 0.513 ** L | 55 |
| Forest ecosystem | 0.597 ** L | 0.624 ** L | 73 |
| Grassland ecosystem | 0.133 ** L | 0.121 ** L | 42 |
| Freshwater wetland ecosystem | −0.051 ** L | −0.025 ** L | 42 |
| Settlement ecosystem | 0.100 ** L | 0.146 ** L | 48 |
| Desert ecosystem | −0.433 ** L | −0.449 ** L | 29 |
| Other ecosystems | −0.261 ** L | −0.266 ** L | 34 |

** $p < 0.01$ (bilateral). L–expressed as linear correlation coefficient, otherwise it is nonlinear correlation coefficient.

From a spatial distribution perspective, the western area of the subtropics and tropics, which belong to the forest ecosystem, are the most abundant in overall number of terrestrial mammals' species (Figures 2 and 8). The region in the northwest extreme arid region of China is classified as a desert ecosystem, and species richness is generally low. The alpine vegetation region of the Qinghai-Tibetan Plateau had the lowest terrestrial mammal richness; this area is a grassland ecosystem, dominated by grasslands, meadows, and alpine vegetation. The mammal species richness in the eastern plains of China also is low, as the area is mainly composed of farmland and settlement ecosystems.

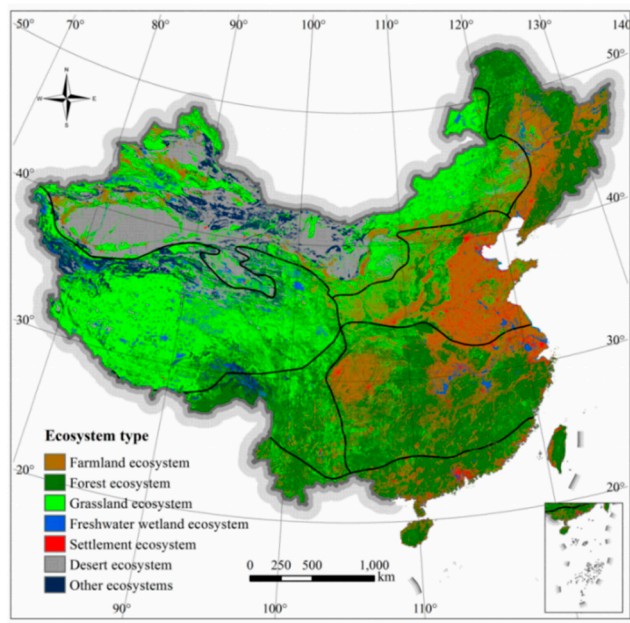

**Figure 8.** Spatial distribution of terrestrial ecosystems in China.

## 4. Discussion

We presented new results on the distribution of mammal species and families richness in China based on GIS spatial analysis. From a data availability standpoint, it was difficult to obtain systematic observation data on animals, which to some extent affects the results of the research. Our study used the latest and most comprehensive species distribution data that can be obtained and utilized at present. Distributions of different mammal species were represented as polygon features that

overlapped with each other in space. Compared with previous regional studies that had been limited to administrative districts, the analysis explored the spatial distribution of terrestrial mammals on the mesh scale to eliminate the influence of area on species richness. Previous studies focusing on the distribution of single species are detailed, but unable to grasp the macro-pattern of mammalian distribution. Our findings may therefore be useful for developing a national strategy for mammal conservation. In the subsequent experiments, we will collect and sort the species-county records, and conduct habitat suitability models in converting county records into grid-based occurrences, to avoid the inflation of species geography ranges and improve the accuracy of species distribution data.

Species richness and family richness were highest in southern China. One reason for the high richness in the south could be because this region is a suture zone between major global biogeographic realms (See [59]). These regions have a mild climate and precipitation regimes, and dense forests that are suitable habitat for mammals. Especially the Yunnan Province in southwest China, due to low latitude, many differences in elevation, complex and diverse landforms, small temperature differentials, and distinct dry and wet seasons are prevalent, which provides good places for many animals to live and reproduce and brings together different types of animals, and thus species richness is rich.

Species richness in the northwest extreme arid region of China is generally low. That region has low precipitation, as well as ongoing desertification and soil erosion. The climatic regime is not suitable for most mammals, only a few of which can adapt to the prevailing ecological conditions.

Areas of relatively low species richness of terrestrial mammal are mainly located in eastern and western China. It is surrounded by mountains in western China, a unique alpine plateau climate has developed due to its complex topography. The environment is characterized by high terrain regions, very cold conditions, and drought climate, with large temperature differences between day and night, and otherwise harsh ecological conditions. Many natural grasslands on the plateau have been destroyed or deteriorated by overgrazing [60,61], resulting in a relatively low, albeit unique, mammal richness.

In eastern China, the terrain is wide and flat, and rich in water, which are suitable for the development of agriculture and livestock husbandry. Human population is dense in cities, resulting in a long-term pattern of deforestation and forest destruction. It is possible that the impact of human activities have had substantial effects on the environment, the environment of mammals may be affected because of highly developed industrial and agriculture regions in the area.

## 5. Conclusions

The distribution of species richness formed four grouping patterns, being high in southern China, low in northern China, and two low-value subsidence areas in eastern and western China. South China has rich animal resources. Particular efforts should be made to protect habitats wherein reside national protected wild plant and animal species, and promoting the establishment of biological corridors among protected areas. The network of protected areas should be enlarged by establishing and integrating nature reserves according to ecological functional areas, and restoration efforts for of ecosystems in northern China should be strengthened. Scientifically-based networks of nature reserves are established in western China. In addition, the protection of endemic and rare species populations and their habitats need to be strengthened. With the rapid economic development in eastern China, it is crucial to establish in that region nature reserves, small reserves, and protected sites, that may focus on the protection of key remaining animals and plants, and to strengthen the preservation of the flora and fauna in densely populated areas of eastern China.

By comparing the species richness distribution of different orders, the distribution rules of different groups among terrestrial mammals were explored from multiple perspectives. Mammals in the orders Perissodactyla, Pholidota, Scandentia, and Proboscidea are few and sparsely distributed in China, including critically endangered species; protection of their living environment should be strengthened. We showed that patterns of spatial distribution of species richness in mammals of China are the result of multiple factors. Humans have strongly affected the distribution pattern of species, thus measures

should be taken to limit and reduce the influence of these adverse factors on animals. Knowledge of the distribution of mammal species will result in improved human–animal interactions. Understanding these spatial patterns can help to make policy for conservation and sustainable development, allowing researchers to pinpoint areas that are in particular need of conservation.

**Author Contributions:** Y.C. conducted the primary experiments, cartography, and analyzed the results. J.W. provided the original idea for this paper. C.X., T.Q., and C.S. actively participated throughout the research process and offered data support for this work. All authors have read and agreed to the published version of the manuscript.

**Funding:** This research was funded by the National Natural Science Foundation of China, grant number 41871294.

**Acknowledgments:** We would like to express appreciations to colleagues in the laboratory for their constructive suggestions. Also, we thank the anonymous reviewers and members of the editorial team for their constructive comments.

**Conflicts of Interest:** We declared no potential conflicts of interest with respect to the research, authorship, and/or publication.

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
