# Peer review of "Spatial Pattern of Species Richness among Terrestrial Mammals in China"

_diversity, doi:10.3390/d12030096_

Round 1
Reviewer 1 Report
The authors address a very interesting and highly relevant research topic by quantifying biodiversity values in a spatially explicit manner.
The research presented in this paper is fully based on the use and analysis of species distribution data taken from Jiang et al. (2015). However, this data is not described in enough detail to understand what the results mean; species distribution used as input may represent species potential habitat data, or species ranges, for example. Is species distribution based on species observations? Or on the presence of suitable habitat (based on what underlying information?)? Do the maps in Figure 2 and 3 represent actual species richness or potential species richness?
The maps of species richness are then linked (amongst other factors) ecosystem types, which includes i.a. farmland, forest and grassland. No source is given for the ecosystem type data, which makes it difficult to assess underlying assumptions of this data, which were not provided by the author.
If, for instance, the distribution data presented are consisting of range maps, species may not occur in all areas within that range (depending on habitat suitability/availability within the range), and species richness will be overestimated if range data for different species is added up to provide maps of species richness. If the distribution data is based on habitat suitability, then this probably also includes some measure of suitability based on land use, which may or may not be based on the same source used in the manuscript to determine species richness per ecosystem type.
Because this information is not provided, Im unfortunately not able to provide a review of the analysis or results. The authors should provide more information on the underlying assumptions of the input data to allow for a detailed review.
Author Response
Thank you for the reviewers’ comments concerning our manuscript entitled “Spatial pattern of species richness among terrestrial mammals in China” (ID: diversity-707752). Those comments are all valuable and very helpful for revising and improving our paper, as well as the important guiding significance to our researches. We have studied comments carefully and have made correction which we hope meet with approval. Here we responded to the reviewers’ comments are as follows. The main corrections are in the manuscript and marked in red in revised portion. These changes will not influence the framework of the paper.
[Comment 1] The research presented in this paper is fully based on the use and analysis of species distribution data taken from Jiang et al. (2015). However, this data is not described in enough detail to understand what the results mean; species distribution used as input may represent species potential habitat data, or species ranges, for example. Is species distribution based on species observations? Or on the presence of suitable habitat (based on what underlying information?)? Do the maps in Figure 2 and 3 represent actual species richness or potential species richness?
[Response] Thank you for your very constructive comments. There is no systematic collection of China’ mammalian distribution data, especially for endangered species. In the study, mammal species’ distribution data were obtained from China’s Mammal Diversity and Geographic Distribution (Jiang et al. 2015), which is the outcome of collective wisdom. Jiang et al. (2015) used A Completed Checklist of Mammal species and subspecies in China: A Taxonomic and Geographic Reference by Wang Yingxiang (2003) and Mammal Species of the World (3rd Edition) by Wilson and Reeder (2005) as templates, collected all available relevant information of books and literatures on taxonomy, distribution of mammals in China by the end of March of 2015. After five evaluation meetings and two rounds of evaluation by correspondence, with input from about 80 mammalogists in China, they finalized The Checklist of China Mammal Diversity. Thus, the most updated information on China's mammal diversity is presented in this book, the distribution of each species is plotted. The species distribution maps are compiled from national and regional atlases and books, as well as on species observations of field work. The maps in Figure 2 and 3 represent actual species richness. This data of species distribution in this paper used as input represent species ranges. We made some changes in “2.2.Data resources” marked in red in the paper.
[Comment 2] The maps of species richness are then linked (amongst other factors) ecosystem types, which includes i.a. farmland, forest and grassland. No source is given for the ecosystem type data, which makes it difficult to assess underlying assumptions of this data, which were not provided by the author.
[Response] Thank you for valuable comments. The macro structure data of China's land ecosystems are based on the data of 1: 100,000 land use/land cover obtained by remote sensing interpretation. Through the identification and research of each ecosystem type, the spatial distribution dataset of terrestrial ecosystem types in China was formed after classification processing. The data set is provided by Data Center for Resources and Environmental Sciences, Chinese Academy of Sciences (RESDC) (http://www.resdc.cn). Ecosystem type has 7 categories, including farmland ecosystems, forest ecosystems, grassland ecosystems, water and wetland ecosystems, desert ecosystems, settlement ecosystems, and other ecosystems. These data are preprocessed before putting into use, including clipping and transforming uniform Asia Lambert Conformal Conic projection. We made some changes in “2.2.Data resources” marked in red in the paper.
[Comment 3] If, for instance, the distribution data presented are consisting of range maps, species may not occur in all areas within that range (depending on habitat suitability/availability within the range), and species richness will be overestimated if range data for different species is added up to provide maps of species richness. If the distribution data is based on habitat suitability, then this probably also includes some measure of suitability based on land use, which may or may not be based on the same source used in the manuscript to determine species richness per ecosystem type.
[Response] Thank you very much for your very constructive comments. There is incomplete information and data of mammal distribution in China, we used the most updated information and distribution data on China's mammal diversity taken from Jiang et al. (2015). The distribution data presented are consisting of range maps in this paper. We analyzed the present situation of terrestrial mammalian distribution in China as a whole from the macro level and compared the aggregation range of species distribution. Then, we mainly measure the influence of specific factors on species richness from a macro perspective. Distributions of different mammal species were represented as polygon features that overlapped with each other in space. The analysis explored the spatial distribution of terrestrial mammals on the mesh scale to eliminate the effect of area on species richness. We didn't take into account the fact that adding up the range data of different species would overestimate species richness. In the subsequent experiments, to further improve the accuracy of animal distribution data, we will collect and sort the species-county records, and will conduct habitat suitability models in converting county records into grid-based occurrences. We think this is a lot of preliminary work. We made some changes in “4.Discussion” marked in red in the paper.
Once again, thank you very much for your constructive comments and suggestions which would help us in depth to improve the quality of the paper. Please see the attachment.
Yours sincerely,
Corresponding author:
Name: Jiechen Wang
E-mail: jiechen_wang@outlook.com

Reviewer 2 Report
Review report on Diversity- manuscript ”Spatial pattern of species richness among terrestrial mammals in China”
The paper describes the spatial patterns of mammal richness in 10 x 10 km grids in China and found areas of higher and lower species richness. High richness was associated to southern locations and this pattern was obvious for most of the species groups (order or family).
The subject was interesting and useful description of mammal fauna in the whole China. I did not find any major flaws from the manuscript but the presentation needs clarification and improvement in several parts of the ms.
L17. Is there three or four agglomerations? At least the authors have listed four.
L48. Delete “ Y”.
L82. “…more systematic…” This is more systematic than what?
L96 Define “DEM”.
L100-104. Is this related to the “altitudinal amplitude” which was first mentioned in L253? If it so, the term should be mentioned already here.
L127. It is not clear how the correlation between richness and ecosystem type was calculated. Does it involves percentage/proportion of a certain ecosystem type within a 10 x 10 km cell? Please, clarify.
Results and analysis and Discussion. Result section should include only pure results and avoid the interpretation of results. Interpretation should be in Discussion. For example, L142-145 has text, which should be in discussion. Discussion starting from line 236 contains mostly results. These results should be moved under the Results and analysis.
L215. Level of threat to species. This was not at all mentioned in introduction or in methods. It comes as a surprise. This should be mentioned earlier. There is repetition in the first paragraph of this section. Delete unnecessary.
L243. There are no results to support this sentence. Can be deleted.
General. Diversity, species number and species richness are all used. Use should be consistent throughout the text.
Text needs linguistic revision.
Author Response
Thank you for the reviewers’ comments concerning our manuscript entitled “Spatial pattern of species richness among terrestrial mammals in China” (ID: diversity-707752). Those comments are all valuable and very helpful for revising and improving our paper, as well as the important guiding significance to our researches. We have studied comments carefully and have made correction which we hope meet with approval. Here we responded to the reviewers’ comments are as follows. The main corrections are in the manuscript and marked in red in revised portion.
[Comment 1] L17. Is there three or four agglomerations? At least the authors have listed four.
[Response] Thank you for your very constructive comments. There is four agglomerations in this paper, high richness in the south, low in north, and two low richness areas in eastern and western China. We have modified it in L16 of this paper.
[Comment 2] L48. Delete “ Y”.
[Response] Thanks for your valuable suggestion. We have deleted "Y" in line 48.
[Comment 3] L82. “…more systematic…” This is more systematic than what?
[Response] Thank you for the comments. About data resources. We checked the GBIF database before. Many data of the GBIF may be easily observed and acquired like birds and butterflies. In terms of observational data among animals, GBIF database have provided relatively abundant observation data with a long time span. However, there is no systematic collection of mammalian distribution data, especially for endangered species. In the study, mammal species’ distribution data were obtained from China’s Mammal Diversity and Geographic Distribution (Jiang et al. 2015), which is organized by many zoologists. We think the data we used are more systematic. We made some changes in “2.2.Data resources” marked in red in the paper.
[Comment 4] L96 Define “DEM”.
[Response] Thank you for your very constructive comments. We have defined “DEM”--Digital Elevation Model in line 107.
[Comment 5] L100-104. Is this related to the “altitudinal amplitude” which was first mentioned in L253? If it so, the term should be mentioned already here.
[Response] Thank you for your very constructive comments. Yes, L100-104. is related to the “altitudinal amplitude”. To classify topographic fluctuation, we defined altitudinal amplitude. We use “altitudinal amplitude” consistently throughout the text. We made the changes in the places marked in red in the paper.
[Comment 6] L127. It is not clear how the correlation between richness and ecosystem type was calculated. Does it involves percentage/proportion of a certain ecosystem type within a 10 x 10 km cell? Please, clarify.
[Response] Thank you for your comments. We count the every ecosystem types in each 10 by 10 km blocks respectively. Then, we used correlation analysis on all cells covering the same ecosystem type and their corresponding species richness.
[Comment 7] Results and analysis and Discussion. Result section should include only pure results and avoid the interpretation of results. Interpretation should be in Discussion. For example, L142-145 has text, which should be in discussion. Discussion starting from line 236 contains mostly results. These results should be moved under the Results and analysis.
[Response] Thank you for the comments and we have carefully edited the manuscript. We made some changes in the manuscript sections, including Results and analysis, Discussion, and Conclusion. Result section should include only pure results and avoid the interpretation of results. The interpretation in the original Results section has been moved to the Discussion. For example, L142-144 has been in discussion. The results of the original Discussion has been placed in the Results and analysis section. More detail was added in the Results and analysis section, such as 3.4.Factors affecting terrestrial mammalian distribution in China. Results and Discussion are not combined. We made the changes in the places marked in red in the paper.
[Comment 8] L215. Level of threat to species. This was not at all mentioned in introduction or in methods. It comes as a surprise. This should be mentioned earlier. There is repetition in the first paragraph of this section. Delete unnecessary.
[Response] Thank you for the comments. We added the sentence in introduction: “We also analysed conservation status of mammals and distribution of threatened species”. We deleted some unnecessary and repeated sentences: “The rhinos like Rhinoceros unicornis, Rhinoceros sondaicus and Dicerorhinus sumatrensis have become Regionally Extinct with no records in China more than 60 years”. We made the changes in the places marked in red in the paper.
[Comment 9] L243. There are no results to support this sentence. Can be deleted.
[Response] Thank you for the comments. We have modified this sentence in the paper: “The distribution of species varies and is latitude dependent”. We made the changes in the places marked in red in the paper.
[Comment 10] General. Diversity, species number and species richness are all used. Use should be consistent throughout the text.
[Response] Thank you for your very constructive comments. We analyzed the distribution of species richness in the paper, using “species richness” in order to ensure the consistency throughout the text. When need to specify the number of species, we used “species number”. “Species diversity” is mentioned when references are cited. We made some changes in the places marked in red in the paper.
Once again, thank you very much for your constructive comments and suggestions which would help us in depth to improve the quality of the paper. Please see the attachment.
Yours sincerely,
Corresponding author:
Name: Jiechen Wang
E-mail: jiechen_wang@outlook.com

Reviewer 3 Report
This is a clear and straight-forward description based on the best data available. Make corrections as indicated.

Author Response
Thank you for the reviewers’ comments concerning our manuscript entitled “Spatial pattern of species richness among terrestrial mammals in China”. Those comments are all valuable and very helpful for revising and improving our paper, as well as the important guiding significance to our researches. We have studied comments carefully and have made correction which we hope meet with approval. In this revision, we have made some changes combined with your opinions, including adding or removing some words. We have carefully edited the Figure 6 according to the suggestions. We thank you for reminding us that tables 2, 3 and 4 have been modified. The main corrections are in the manuscript and marked in red in revised portion. We tried our best to improve the manuscript. These changes will not influence the framework of the paper.
Once again, thank you very much for your constructive comments and suggestions which would help us both in English and in depth to improve the quality of the paper.
Yours sincerely,
Corresponding author:
Name: Jiechen Wang
E-mail: jiechen_wang@outlook.com
